# An Individualized Multi-Modal Approach for Detection of Medication “Off” Episodes in Parkinson’s Disease via Wearable Sensors

**DOI:** 10.3390/jpm13020265

**Published:** 2023-01-31

**Authors:** Emad Arasteh, Maryam S. Mirian, Wyatt D. Verchere, Pratibha Surathi, Devavrat Nene, Sepideh Allahdadian, Michelle Doo, Kye Won Park, Somdattaa Ray, Martin J. McKeown

**Affiliations:** 1Department of Neonatology, Wilhelmina Children’s Hospital, University Medical Center Utrecht, 3585 EA Utrecht, The Netherlands; 2Department of Electrical Engineering (ESAT), STADIUS Center for Dynamical Systems, Signal Processing and Data Analytics, KU Leuven, B-3001 Leuven, Belgium; 3Pacific Parkinson’s Research Centre, Djavad Mowafaghian Centre for Brain Health, University of British Columbia, Vancouver, BC V6T 2B5, Canada; 4Clinical Fellow-Neurophysiology, Columbia New York Presbyterian, New York, NY 1032, USA; 5Department of Medicine, Division of Neurology, The University of Ottawa, Ottawa, ON K1Y 4E9, Canada; 6Department of Neurology, Penn State Milton S. Hershey Medical Center, Hershey, PA 17033, USA; 7Faculty of Medicine (Neurology), University of British Columbia, Vancouver, BC V6T 2B5, Canada

**Keywords:** wearable, biomarkers, wearing-off, Parkinson’s disease, empirical mode decomposition, canonical correlation analysis

## Abstract

The primary treatment for Parkinson’s disease (PD) is supplementation of levodopa (L-dopa). With disease progression, people may experience motor and non-motor fluctuations, whereby the PD symptoms return before the next dose of medication. Paradoxically, in order to prevent wearing-off, one must take the next dose while still feeling well, as the upcoming off episodes can be unpredictable. Waiting until feeling wearing-off and then taking the next dose of medication is a sub-optimal strategy, as the medication can take up to an hour to be absorbed. Ultimately, early detection of wearing-off before people are consciously aware would be ideal. Towards this goal, we examined whether or not a wearable sensor recording autonomic nervous system (ANS) activity could be used to predict wearing-off in people on L-dopa. We had PD subjects on L-dopa record a diary of their on/off status over 24 hours while wearing a wearable sensor (E4 wristband^®^) that recorded ANS dynamics, including electrodermal activity (EDA), heart rate (HR), blood volume pulse (BVP), and skin temperature (TEMP). A joint empirical mode decomposition (EMD) / regression analysis was used to predict wearing-off (WO) time. When we used individually specific models assessed with cross-validation, we obtained > 90% correlation between the original OFF state logged by the patients and the reconstructed signal. However, a pooled model using the same combination of ASR measures across subjects was not statistically significant. This proof-of-principle study suggests that ANS dynamics can be used to assess the on/off phenomenon in people with PD taking L-dopa, but must be individually calibrated. More work is required to determine if individual wearing-off detection can take place before people become consciously aware of it.

## 1. Introduction

Parkinson’s disease (PD) is a multisystem disease associated with both motor and nonmotor aspects [1,2]. The main biochemical abnormality in PD is the lack of dopamine [3], so the dopamine precursor levodopa (L-dopa), which crosses the blood-brain barrier, is the mainstay for treatment. In advanced disease stages when the brain loses its buffering capacity, the patients may feel “ON” and “OFF” throughout the day, where “ON” describes the time when they feel symptoms well controlled after L-dopa intake. When the plasma L-dopa level decreases, symptoms of the disease can re-emerge and the patients may feel “OFF”. The conversion from the “ON” state to the “OFF” state as the plasma L-dopa decays is known as the “wearing-off (WO)” phenomenon [4,5,6]. Because it takes up to an hour for the medication to be absorbed and cross the blood-brain barrier, waiting until WO flares and then deciding to take the next dose is a suboptimal strategy. Taking the medication on a strict dosing schedule is a helpful strategy, but various factors, including delayed absorption, variable metabolism of the medication based on physical activity, and even consumption of protein, can affect how long the effects of a given dose will last. Because WO is physically and mentally uncomfortable and stressful [7], some patients end up consuming excessive quantities of L-dopa to avoid WO. Unfortunately, this can lead to involuntary writhing movements in the on state (“peak dose dyskinesia”) in patients with long disease duration and long-term exposure to L-dopa. People with excessive dyskinesia can end up inexplicably losing weight since they move incessantly [8], often triggering unwarranted investigations. Thus, there is considerable interest in determining a biomarker for predicting WO before it becomes overt.

There are strong reasons to believe that early prediction of WO might provide not only immediate benefits but also affect the long-term outcome of the disease in a positive way. The “continuous dopamine stimulation hypothesis” [9,10] posits that fluctuations in plasma levels of dopamine are a contributing factor to the development of subsequent dyskinesias. In fact, L-dopa/Carbidopa Intestinal Gel therapy is based on the continuous infusion of L-dopa gel into the jejunum to get as smooth a serum profile of L-dopa as possible [11] by bypassing the stomach. By preventing OFFs, there might be better control of motor fluctuations in the long term [12].

A number of studies have attempted to detect L-dopa levels from sweat in healthy volunteers. The L-dopa concentration in three subjects after fava bean consumption and active exercise was estimated via iontophoresis contained within a wearable strap [13]. Sweat from a finger touch-based method was able to measure L-dopa levels that correlated with capillary blood samples [14]. A microneedle sensor in a skin-mimicking phantom gel has also been shown to detect L-dopa in continuous fashion [15]. However, to our knowledge these technologies have yet to be explored in actual subjects with PD.

Most work related to the prediction of WO in PD has been with accelerometer data [16,17,18,19,20,21]. However, there are potential issues with this approach. It can be difficult to discern volitional movement from other pathological movements seen in PD, such as tremor and dyskinesia, especially when only a single wrist-based sensor is used. Discrimination between normal and pathological movements can be more easily obtained by using multiple sensors across the body, but in that case compliance is poor: many people with advanced disease are not amenable to wearing multiple sensors for prolonged periods. Finally, the ultimate goal would be the capability of predicting WO before people become consciously aware that they have worn-off. This is in accordance with prior research on the relation between some biomarkers of the autonomic nervous system (ANS) and early warning signs of WO episodes [22,23,24]. Presumably, a reduction in overall movement, as assessed via accelerometers, would likely be noticed by the subject. 

A number of ANS markers are suitable for mobile monitoring. Time-varying electrodermal activity (EDA) signals and heart rate (HR) are modulated by physical, orthostatic, and cognitive stress that are known to activate sympathetic tone [25]. Blood volume pulse (BVP) ambulatory monitoring is feasible and may be an important marker for WO, as hypotensive episodes are significantly reduced after the continuous infusion of L-dopa [26]. Alterations in skin temperature (TEMP) (i.e., skin TEMP) may be an important marker for thermoregulatory dysfunction common in PD [27].

Here, we aimed to test the hypothesis that ANS markers (EDA, HR, BVP, and skin TEMP) provided by a single commercial wrist sensor (E4 wristband^®^) could serve as a tool for developing a non-invasive biomarker for WO episodes. We decomposed the aforementioned signals into different waveforms using Empirical Mode Decomposition (EMD). This method takes a given signal and separates it into smoother and smoother “Intrinsic Mode Functions (IMFs)”. A key benefit of this approach, unlike more common Fourier methods, is that it can deal with non-stationary signals, i.e., signals where the frequency components vary over the duration of an epoch. We demonstrate that feature extraction by the empirical mode decomposition (EMD) of multimodal E4 sensor signals can help to predict the moments when PD subjects feel the need for medication (i.e., the OFF state) in an individualized fashion.

## 2. Materials and Methods

### 2.1. Study Design and Participants

This prospective study was carried out at the Pacific Parkinson’s Research Center, University of British Columbia. Following approval by the Ethics Board, all subjects had provided written, informed consent. 

We recruited 25 patients diagnosed with PD by certified movement disorder specialists according to the United Kingdom Parkinson’s Disease Society Brain Bank Criteria. Exclusion criteria included (i) atypical Parkinsonism, (ii) depressive mood identified by Beck Depression Inventory-II (BDI-II)>14 or concurrent treatment with antidepressants, (iii) cognitive impairment measured by Montreal Cognitive Assessment ≤ 22, (iv) history of epilepsy, polyneuropathy, spinal cord diseases, thyroid dysfunction, or severe dermatological conditions, and (v) history of deep brain stimulation, implantation of any medical devices, or anticholinergic medication use. 

Demographic features including age, sex, duration of disease after initial diagnosis, and total daily L-dopa dose were obtained. Overall severity of Parkinsonism was assessed by Unified Parkinson’s Disease Rating Scale (UPDRS) part III in all participants. The “wearing off questionnaire-19 (WOQ-19)”, a clinical scale that measures the degree of fluctuations in both motor and non-motor symptoms, was assessed in all patients. A compact view of subjects’ responses to the questionnaire is plotted in Appendix A. Participants were considered to have WO if they experienced at least two or more symptoms in WOQ 19 that improved with L-dopa intake [28,29]. The Scales for Outcomes in Parkinson’s disease-AUTonomic dysfunction (SCOPA-AUT) was used to assess any pre-existent autonomic system function [30]. Table 1 lists the main demographic features of each subject.

A wearable wristband, E4 wristband^®^ (Empatica Inc., Milan, Italy), was used to obtain EDA, HR, BVP, and TEMP information. It carries out EDA measurements with two dry silver-plated electrodes that are attached to the inner surface of the watch with a sampling rate of 4 Hz and a range of skin conductance (SC) from 0.01 μS to 100 μS. BVP is measured by photoplethysmography (PPG) (and from that, HR can be inferred) [31]. The PPG sensor evaluates the volume of the passing blood along tissues using a photo-detector that computes the reflection coefficient of light (generated by PPG’s light source) from the skin [32]. The sampling rate of BVP is 64 Hz. The E4 is also equipped with an infrared thermopile sensor that reads peripheral skin temperature (with a frequency sampling rate of 4 Hz) [33].

An activity log was kept by the patients that contained their self-reports of time past from the latest dose, sleep, and ON/OFF reports performed every 30 minutes while wearing the device for a period of 24 hours. A report of “OFF” from any patient meant that he/she felt the urge for the medication, and “ON” had the opposite interpretation. They were instructed to carry out normal daily activities while being careful not to dislodge the electrodes and avoiding exposing the device to water. The E4 data were downloaded from the device later offline. For this proof-of-principle study, we restricted ourselves to the 12 subjects who documented “OFF” states in their diary. The other 13 subjects declared they had WO episodes but had failed to record this in their diary.

### 2.2. Method

The schematic view of the analysis approach is shown in Figure 1. Each patient’s data of EDA, HR, TEMP, and BVP signals were divided into 30 s epochs, resulting in 2880 epochs over 24 hours. Since the subject ON/OFF report from each patient was every hour, and the sensor-based recordings were available in 30 s epochs, we up-sampled and interpolated the ON/OFF data to create an approximate label for every 30 seconds. This resulted in a gradual transition between ON/OFF states, as observed in Appendix A. The sensor signals were decomposed using the empirical mod decomposition (EMD) method, as explained in Section 2.2.1. EMD was then applied on the four sensor signals in order to extract features that could potentially show a correlation with up-sampled ON/OFF data. We then used different train-test strategies to assess the degree of sensitivity of the ON/OFF-related biomarkers we obtained for each individual participant (described below)**.**

#### 2.2.1. Pre-Processing and Feature Extraction

The ON/OFF signals were constructed by up-sampling the hourly ON/OFF designation to have one ON/OFF value every 30 seconds. The resampling was carried out linearly with an FIR anti-aliasing lowpass filter with a Kaiser window [34] equal to 5 and a neighbor term number of 10. (An example of one of the up-sampled label signals is shown in Appendix A.) Here, we assume that the target signal should be relatively smooth over the adjacent epochs.

We used EMD to decompose the non-stationary multimodal signals from the E4 sensor into intrinsic mode functions (IMFs) [34]. EMD decomposes the original signal from its higher frequency components to lower ones [35], halting when the final residual reaches a termination criterion. However, it is not bound to the specific single frequency content of each dynamic basis as the Fourier method is. Therefore, there is no presupposition of the linearity and time invariance of the physiological signal generation by human organs. Therefore, each 1-D signal for 24 hours is decomposed into multiple 1-D time dynamics that have different frequency components and improve feature extraction ability through an increase in the dimensionality of our data. 

The resultant intrinsic mode functions (IMFs), along with the residuals, are the complete expansion of the original multi-component signal and are also nearly orthogonal [36]. The fact that the number of extrema and zero crossings of IMFs can be different by at most one helps to attain nearly sinusoidal oscillatory components of the original signal. Each IMF is constructed by interpolating between the envelopes of the minimum and maximum extrema of the original (or what has left after consecutive subtraction of IMFs from the original) signal in a way that the mean value of these two constructed envelopes should be near zero (or ideally becomes zero) [35].

In this paper, EMD of each of the four aforementioned sensor signals was calculated by the MATLAB in-built function “emd” using the stop criterion “Maximum Original to Residual Energy Ratio” set to 30, the “Maximum number of IMFs’’ set to 15, and the “Minimum Number of Extrema in Residual Signal” set to zero. The “Sifting Relative Tolerance” was set to 0.1, and the envelope interpolation was conducted by a spline method. 

Although a termination criterion can be used for EMD, we empirically found that the first 15 IMFs contained the most useful information. For each IMF, we computed the mean, standard deviation (STD), and entropy values of each epoch [37] for EDA, HR, BVP, and TEMP signals. Thus, for each epoch, we had {15 IMFs × 4 modalities × 3 statistics} = 180 features to be used for subsequent classification. Thus, the dimension of the feature matrix (X) for each participant’s data was at most 2880 epochs by 180 features for each individual subject.

The joint features were anticipated to provide the most accurate prediction of WO. However, to see whether or not each of the sensor signals individually could also lead to adequate WO prediction, we also separately decomposed only one signal from EDA, TEMP, BVP, or HR at a time for comparison. 

To determine if the WO features were individually specific, we compared the performance of models based on pooling the data across subjects vs. testing each subject individually.

#### 2.2.2. Intersubject Approach

First, we pooled all the data across all subjects. Each signal was recorded for 24 h, and features were extracted for every 30-second epoch. Thus, there were 12*2880 epochs. The CCA algorithm was trained on 65% of the data (randomly chosen) and validated on the remaining 35%.

#### 2.2.3. Subject-Specific Approach

We trained the model on a random split of 65% of each subject’s data separately and tested the regression canonical-correlation analysis (CCA) coefficients on the remaining 35% (resulting in different weights for each subject). So, for each subject, 1872 and 1008 epochs were utilized as test and train data sets, respectively. 

Finally, to ensure that the subject-specific approach did not result in overfitting, we fed in a random signal (noise generated by MATLAB rand function) as a test. This was to ensure that the high correlation between the reconstructed and true OFF state was not a spurious response and not due to model overfitting. Result is depicted in Appendix A.

## 3. Results

### 3.1. Intersubject Approach

Figure 2 depicts the original Y (upsampled OFF data recorded from patients) and the reconstructed Y from the models, based on sensor signals. The data have been concatenated from all subjects, resulting in 260 h (12 subjects × 24 h). Qualitatively, one can observe that an individualized approach is fairly successful in predicting OFF states across subjects. The correlation with Y of the training data is 69%, and it is 68% for the test data (*p*-value < 1 × 10^−10^). Note that the “OFF” state is considered positive on this axis.

### 3.2. Subject-Specific Approach

Figure 3 depicts the correlation of the true OFF and the reconstructed OFF obtained by a linear combination of extracted features from the four sensor signals in the analysis for both training and testing data.

In general, the correlation between both training and test data was >0.9. A detailed comparison between the reconstructed OFFs and true upsampled OFFs for each subject are described in the Appendix A.

To determine the most important features for each subject, we plotted features having weights above 50% of the maximum magnitude coefficient for representative subjects (Figure 4). Through these subplots, we can see which features and from which signal contribute most to the prediction of OFF states. It is obvious how variable the dominant features’ ranks are among different subjects. (The full results can be found in the Appendix A.) The features weighted were quite different across subjects (Figure 4).

Table 2 summarize the mean and standard deviation (STD) of correlation between true and reconstructed OFF when each signal used individually. 

In order to test if the use of multimodal signals resulted in better classification than any single modality, we also tested the ability of each single modality to predict WO (Table 1, Figure 5). Note the relatively poor performance of any single modality to predict WO compared to the combined approach used in Figure 2. Moreover, there is not even a single signal that works best for OFF prediction among all of the subjects.

Although the individual modalities did not perform as well as the combined approach (Figure 2 and Figure 5), the ability of the EDA signal alone to predict WO was weakly correlated with UPDRS part III score (r = 0.62, *p*-value < 0.03) (Figure 6).

Appendix A illustrates the training and test data correlation due to a random noisy signal, showing that the correlation with the fake noise test signal is low, and all have non-significant *p*-values. Appendix A shows the *p*-values of the correlations with the random input signal (as opposed to the true ON/OFF signal) of the same extracted features.

## 4. Discussion

In this study, we demonstrate that the accurate prediction of the WO phenomenon in PD is possible using the subject-specific analysis of ANS sensor recordings. This suggests that a linear, deterministic and individualized relationship exists between ANS recordings obtained from a single wrist-worn sensor and patient-reported OFFs.

Predicting WO (defined as the re-emergence of motor and non-motor symptoms that improve with further L-dopa intake [29]) is a critical aspect of the clinical management of PD. Motor fluctuations can be observed in up to 50–80% of patients with advanced PD [38], and even in the early stage of disease, up to 50% of patients can develop motor fluctuation within the first 2 years of therapy [39]. However, the WO phenomenon is not limited to only motor symptoms (e.g., bradykinesia and tremor); it can span a variety of non-motor symptoms such as anxiety, depression, mood changes, sweating, and other types of discomfort [40]. In such circumstances, there might be a discrepancy between the patient’s reporting of fluctuations and the clinician’s assessment of the OFF state. Studies have reported that patients’ self-reported WO are at a higher frequency than that identified by movement disorder specialists. Overall, WO is a very subjective and heterogeneous phenomenon that varies across patients. Thus, predicting WO on an individual level is a more justifiable approach. This may also explain why the high accuracy at the subject-specific level was not replicated when attempting to implement a “one-size-fits-all” model. 

While we focus here on autonomic markers of WO, the majority of studies thus far have emphasized the use of accelerometers, focusing on the motor aspect of the disease [17,18,19]. However, although less well-known, the prevalence of non-motor fluctuations in patients with PD is very high. The prevalence ranges from 17% to almost 100% [41]. Disruption in autonomic systems was proven in objective measures such as in sweating measurement using an evaporimeter, heart rate variability, and electrodermal activities [42,43,44]. PD patients with motor fluctuations have decreased skin temperature, increased sweating, and higher standing blood pressure in their OFF states compared with in their ON states [43]. Our study suggests the potential of ANS markers to predict WO of PD.

The objective assessment of the disease is being increasingly used for making informed decisions about therapeutic modifications. In recent years, continuous objective measurements enabled by wearable sensors have presented an opportunity to assess motor fluctuations and subsequently guide management [45,46,47]. Wearable technologies also offer possibilities for the quantitative measurements of autonomic impairment. However, sensors to detect autonomic fluctuation in PD in a continuous manner during daily activities have thus-far been rarely investigated (e.g., heart rate variability (HRV) in PD [48]). Figure 5 clearly illustrates how subject-specific the different modalities are in predicting WO. While EDA does appear to be the best ANS marker for WO prediction, this was not the case for all subjects (such as in subject no. 4, 7, 8, and 12). This again emphasizes the need for individualized models specific to each patient in the future. 

Our results, which indicate the need for individualizing the optimal combination of autonomic features to predict WO, are likely a direct consequence of the fact that WO symptoms themselves are highly heterogeneous. We note that the original questionnaire to assess WO (which, as with all questionnaires, attempts to prevent overlapping questions) included some 32 possible symptoms [49]. In particular, previous descriptions of ANS disturbances during WO have spanned a broad clinical spectrum, encompassing a variety of gastrointestinal, urogenital, sudomotor, respiratory, and cardiovascular symptoms [50]. Such heterogeneity of autonomic symptoms during WO episodes likely arises from the widespread distribution of dopaminergic receptors in both central and peripheral ANS [51] and the variability in the degeneration in these structures. Thus, it may be unrealistic to expect that a “one-size-fits-all” combination of ANS features would be appropriate. However, this implies that to translate our current results into a practical approach, a two-step procedure would be required. One would first need a “calibration phase” to determine which individually specific combinations of ANS markers would be optimal. In this calibration phase, subjects would wear the device for a day, and then the identification of the best autonomic parameter combination for WO prediction would be established. This combination would then be fixed and used longitudinally to predict WO. We cannot discount that a much larger study may detect distinct clusters and/or subtypes of ANS changes with WO. If so, instead of a calibration phase to fully individualize ANS feature combinations as described above, one would just need to select which ANS WO subtype each person belonged to. 

There are a number of limitations to our study. First, a large number of subjects had to be excluded due to not documenting OFF episodes in their activity log during the sensor recording—although they actually reported having OFF episodes during recording when interviewed afterwards. This is perhaps understandable, as patients become more immobilized and apathetic in OFF episodes and are disinclined to fill out a diary. This again emphasized the need for unintrusive and automated assessments of WO. Nevertheless, we cannot discount that there may exist a mismatch between self-reported OFFs and automated assessments that would affect our individualized models. Second, the self-reported OFF episodes do not specify whether the episodes are motor or non-motor worsening and lack objective validation by specialists. The correlation between a patient-filled diary and motor fluctuations captured by a wrist worn sensor has been previously shown to be limited [52]. Studies have shown that a portion of PD patients in fact have impaired subjective awareness of their motor impairments [46,53]. Supplementing objective measurements along with the sensor data and patients’ self-reports would greatly influence the improvement of treatment decisions and outcomes [46]. 

So far, we have demonstrated a deterministic association between subject feelings of OFF and multimodal sensory recordings. However, the next step would need to be to determine if we can predict WO episodes before subjective feelings of WO. Our results would suggest that a two-step approach would be required. One would first need a “calibration phase” to determine which individually-specific combinations of ANS markers would be optimal. During this phase, PD subjects would need to be extra vigilant to accurately mark WO in their diary so that an accurate model could be constructed. After the specific combinations of ANS have been determined, this would be fixed and then used to track WO episodes.

## 5. Conclusions

Using subject-specific models, we were able to show a deterministic relationship between ANS indices detected with a wearable sensor and subjective WO episodes in PD. This proof-of-principle study suggests that ANS markers could be further explored for the monitoring of PD, even in advanced stages.

## Figures and Tables

**Figure 1 jpm-13-00265-f001:**
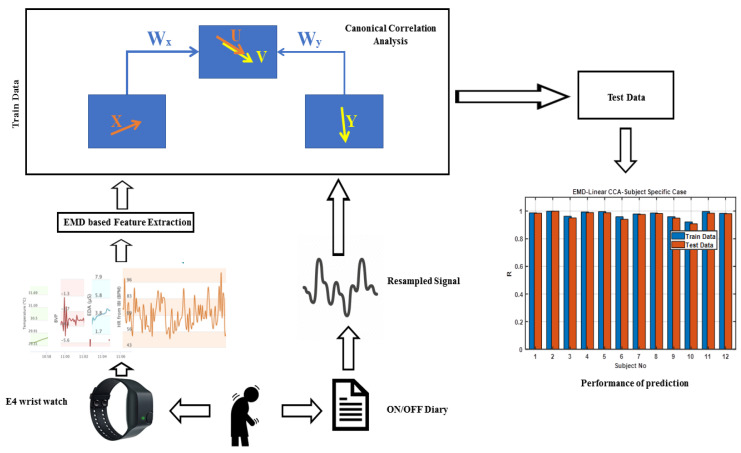
The overview of our proposed method.

**Figure 2 jpm-13-00265-f002:**
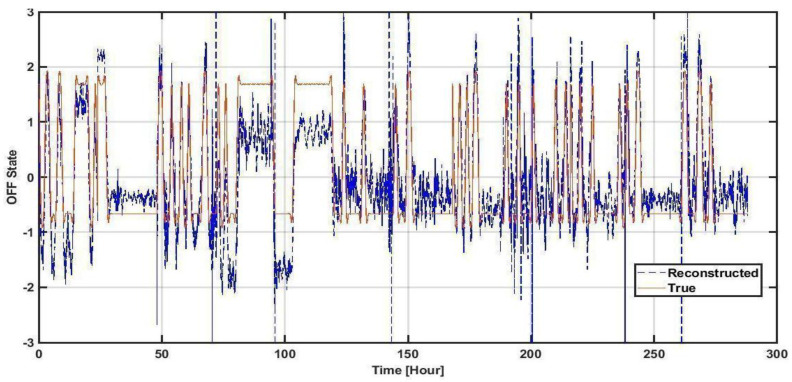
The overview of our proposed method reconstruction of OFF state signal versus true OFF states. On the x-axis, every 24-hour period consecutively comes from a specific subject out of our 12 subjects.

**Figure 3 jpm-13-00265-f003:**
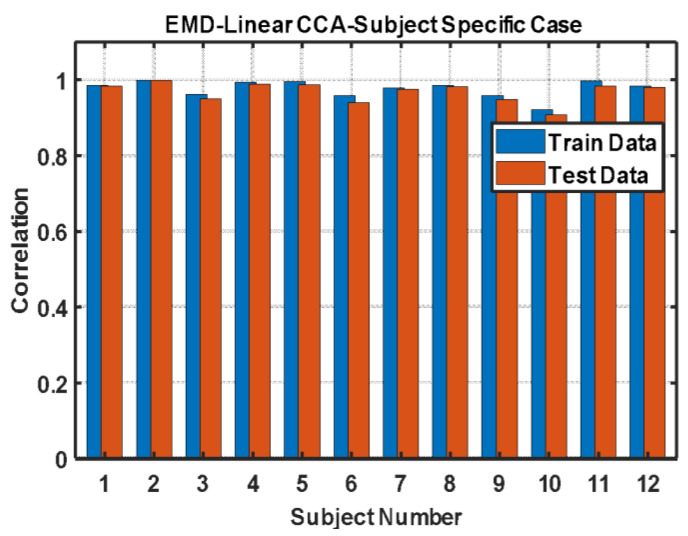
Correlation between the true OFF state and the reconstructed OFF state obtained by the linear combination of the extracted features of four sensor signals.

**Figure 4 jpm-13-00265-f004:**
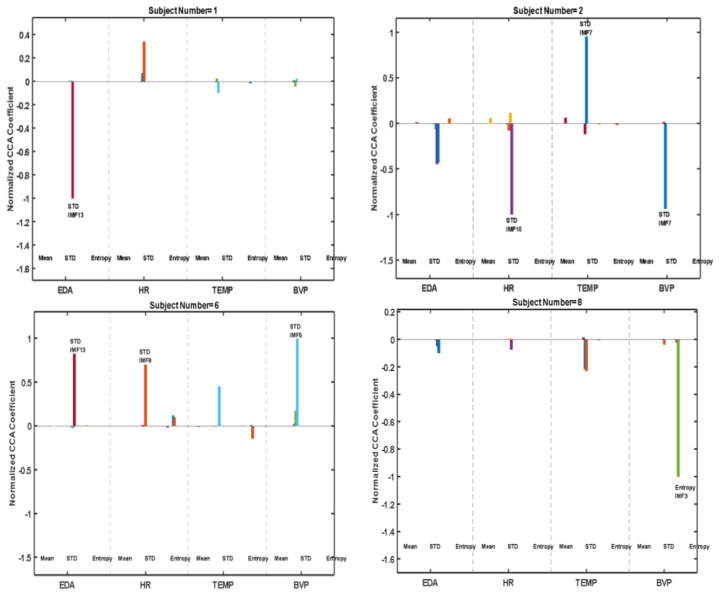
Normalized CCA coefficient weights for each feature were obtained individually for different subjects.

**Figure 5 jpm-13-00265-f005:**
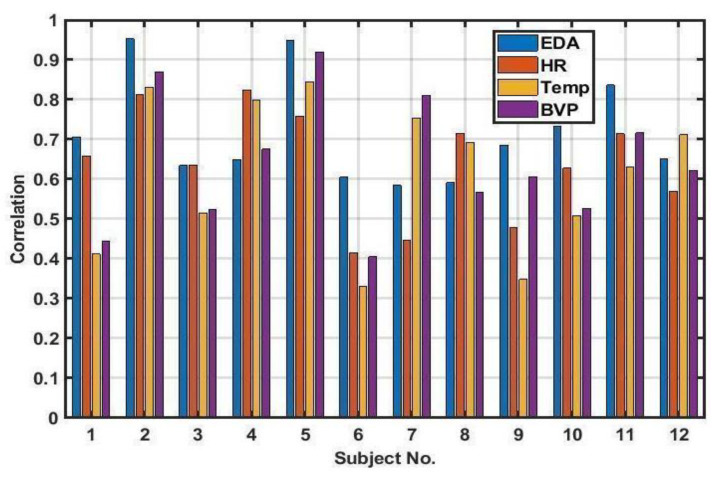
Correlation of reconstructed ON/OFF states derived from any of EDA, BVP, HR, or TEMP signals with the true states. Compare with Figure 3, where each modality is combined in an individually-specific manner to predict WO.

**Figure 6 jpm-13-00265-f006:**
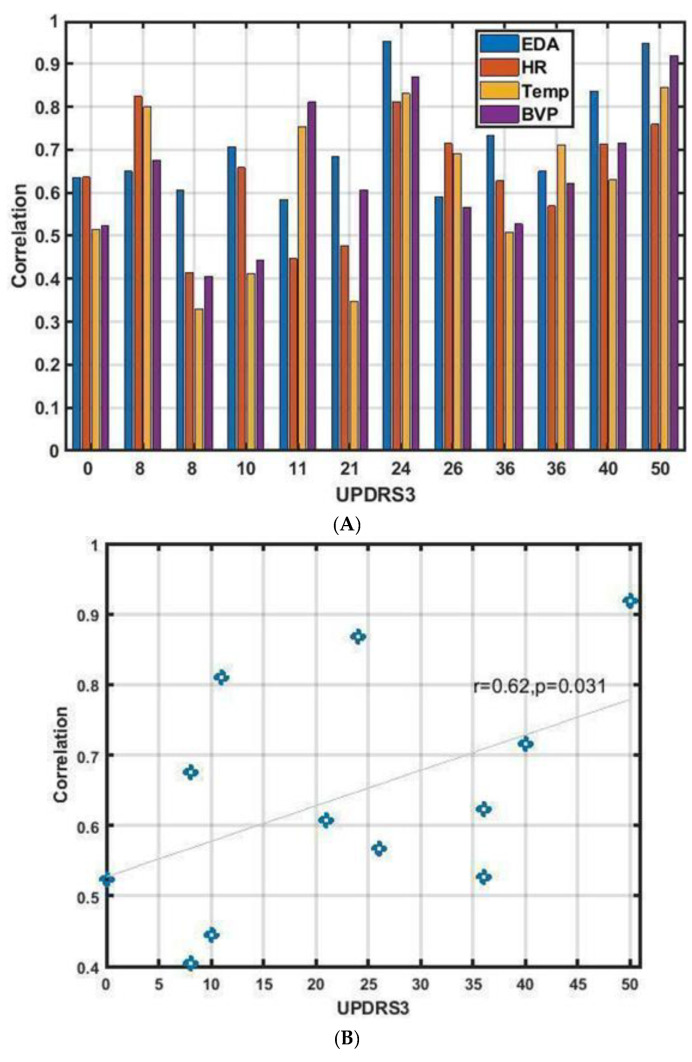
(**A**) Correlation of the true and the reconstructed ON/OFF states obtained by single modalities shown for each disease severity score. (**B**) Correlation of EDA-based reconstructed ON/OFF and UPDRS part III scores. The blue line is the best fitted regression line on the 12 subjects.

**Table 1 jpm-13-00265-t001:** Main demographic features of each subject. BDI: Beck’s Depression Inventory. MoCA: Montreal Cognitive Assessment.

Subject No.	Age (Years)	Sex	Disease Duration (Years)	UPDRS3	MoCA	BDI-II
1	62	Female	3	10	Unavailable	7
2	60	Male	7	24	29	5
3	43	Male	5	0	Unavailable	13
4	64	Male	15	8	29	0
5	72	Male	13	50	28	4
6	74	Female	13	8	27	6
7	36	Male	12	11	30	8
8	72	Male	13	26	29	9
9	64	Female	11	21	29	4
10	54	Male	6	36	30	8
11	67	Female	4	40	26	6
12	58	Male	6	36	22	14

**Table 2 jpm-13-00265-t002:** Comparison of each sensor signal’s effect in forming maximum possible correlated features with OFF state signal for subject-specific analysis (correlation computed between test data of original OFF state and the reconstructed Off (averaged over all 12 subjects)).

Sensor Signal	Mean and STD of Correlation
EDA	Mean = 0.71, STD = 0.14
HR	Mean = 0.57, STD = 0.15
TEMP	Mean = 0.51, STD = 0.26
BVP	Mean = 0.65, STD = 0.17

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
