# Peer review of "An Individualized Multi-Modal Approach for Detection of Medication “Off” Episodes in Parkinson’s Disease via Wearable Sensors"

_jpm, 2023, doi:10.3390/jpm13020265_

Round 1
Reviewer 1 Report
Dear Authors
The article is interesting but there are some limitations which need to be addressed and queries to be answered
The number of patients is very small and very heterogenous in terms of disease duration and severity.
Please elaborate on how the assessment data was taken and divided, i.e.,how many hours were eliminated - sleep or excess movements
How were the epochs divided to assess the data (duration of epochs)- how many test epochs were used, how many epochs were used for assessment.
Were there any dyskinetic patients -were these periods eliminated or were they analysed separately
Were the assessments -UPDRS scores (demographic details) taken during 'off' or 'on' period. It would be good if the 'off' scores were only mentioned One of the patients had 0 UPDRS score - that suggests no Parkinson's disease. Two of the other patients had very low scores -8 and 10, which suggests very mild disease- did the sensors help in these patients
Can you mention if the disease duration, severity of the disease, severity of non motor symptoms all had an effect on the assessments. It is a small number of patients but still were there any changes discernable
Author Response
Dear Respected Reviewer,
We are thankful for your great comments. We did our best to answer them comprehensively. We hope you find the responses sufficient and proper. The added senteces to the manuscript are highlighted by green.
Best regards,
Authors.
We have carefully reviewed the comments from the reviewers and addressed the points raised below.
Replies to Reviewer 1
Comment 1:
“The number of patients is very small and very heterogenous in terms of disease duration and severity”.
Response1:
Thank you for your comment. We agree that this is a relatively small sample size for observing a clinical phenomenon. However, our main points in this proof-of-principle study, namely that 1) wearing off is associated with physiological changes that can be detected from wrist-worn sensors and 2) the combination of modalities to best predict wearing off must be individualized, remain valid despite the relatively small sample size. What we can’t say, (and don’t claim in the current manuscript) is whether or not a “one-size-fits-all” recipe for combining modalities would be “almost” as good as an individualized approach for the majority of patients. If that were the case, this would indeed be beneficial, as then an initial “calibration phase” to determine the individualized combinations would not be necessary. However, we will need to defer to a larger study to be performed later, to answer this question.
Comment 2:
“Please elaborate on how the assessment data was taken and divided, i.e.,how many hours were eliminated - sleep or excess movements”
Response2:
Thank you for mentioning that. We based our CCA response input on the reported ON/OFF state by the subject. This may include sleep periods. We believe this is beneficial, as the models that look at what modalities tend to occur together in a given subject may also be relevant during sleep as well as during wake periods. It suggests that the individually-specific combinations of modalities were relatively invariant to the sleep-wake periods and specifically tailored to detect “OFFs” in that individual.
Comment 3:
“How were the epochs divided to assess the data (duration of epochs)- how many test epochs were used, how many epochs were used for assessment”
Response3 :
We agree that this should be made more clearly. The following sentences are added to section 2.2.2. the manuscript to make related details more understandable:
(for inter-subject approach:)“Each signal was recorded for 24 hours and features were extracted for every 30 second epochs. So there are generally 12*2880 time samples (number of 2880 thirty second epochs in 24 hours). The CCA algorithm is trained on 65% of the data (randomly chosen), and validated on the remaining 35%. ”
(for subject-specific approach:) “We trained the model on a random split of 65% of each subject’s data separately and tested the regression canonical-correlation analysis (CCA) coefficients on the remaining 35% (resulting in different weights for each subject). So, for each subject, 1872 and 1008 epochs were utilized as test and train data sets, respectively (each epoch containing 30 second windows). ”
Comment 4:
“Were there any dyskinetic patients -were these periods eliminated or were they analysed separately”
Response 4:
There were a portion of patients (n = 4) who reported dyskinesia episodes while ON during the E4 recording. These periods of dyskinesia were not eliminated or analyzed separately because we aimed to find combinations of modalities that were sensitive to ON/OFF that were relatively insensitive to both sleep/wake state (see above) as well as excessive dyskinesias.
Comment 5::
“Were the assessments -UPDRS scores (demographic details) taken during 'off' or 'on' period. It would be good if the 'off' scores were only mentioned. One of the patients had 0 UPDRS score - that suggests no Parkinson's disease. Two of the other patients had very low scores -8 and 10, which suggests very mild disease- did the sensors help in these patients”
Response 5:
The UPDRS scores were assessed at the time of patients’ visit to the clinic. Thus they were not asked to withdraw medications, and would typically be considered in their “ON” state. The key criteria for inclusion was that they had previously experienced frequent “OFF” episodes.
Comment 6:
“Can you mention if the disease duration, severity of the disease, severity of non motor symptoms all had an effect on the assessments. It is a small number of patients but still were there any changes discernable”
Response 6:
We did observe that higher correlation of the model was associated with higher UPDRS part III score, as shown in Figure 6B. However, in order to minimally impact patients' ongoing care, we did not withdraw medication to assess their UPDRS III score, and assessed them when they presented to the clinic. While our perception is that non-motor OFFs may be predictable with our method, we believe that we are currently under-powered with respect to the number of subjects to be able to determine if non-motor “OFFs” can be predicted to the same degree as motor OFFs.
Reviewer 2 Report
jpm-2132922
Thank you for the opportunity to review the manuscript entitled: An Individualized Multi-Modal Approach for Detection of Medication “Off” Episodes in Parkinson’s Disease via Wearable Sensors
This is a well written manuscript, and a very interesting approach to a real problem in the movement disorders clinic.
However, I have the following comments:
1. In the introduction: “some patients end up consuming excessive quantities of L-dopa to avoid WO. Unfortunately, this can lead to involuntary writhing movements (“dyskinesia”).”
This Idea was very common in the past, however has been revaluated. What we know now is that the dyskinesias are related with the progression of the disease instead of a direct effect of dopamine. I recommend modify this sentence to avoid confusion.
2. “Unconscious biomarkers, e.g., from the autonomic nervous system (ANS), have the potential to be early warnings of impending WO episodes” Please add a reference to proof this sentence. Or is it the hypothesis of this study?
3. “are modulated by known activators of sympathetic tone, such as cold pressor tests, head-up tilt, stand test, and Stroop tests”
Please clarify this sentence. The Stroop test is a cognitive test, and according with the cited reference, it was used as “Cognitive stress” to increase sympathetic activity. It can create confusion in a reader that is unfamiliar with this topic.
4. We demonstrate that feature extraction by empirical mode decomposition (EMD) of multimodal E4 sensor signals can help to predict the time when PD subjects feel the need for medication (i.e., the OFF state) in an individualized fashion
Please clarify this sentence. The readers of this journal are from different disciplines and no familiarized with EMD. Additionally, this conclusion has not been proved totally, since this is a proof of principle study and its small sample
does not allow to do a generalization.
5. There is no mention of the results of the patients self-report questionnaire (WOQ-19).
6. “However, although less well known, the prevalence of non-motor fluctuations in patients with PD is very high. The prevalence ranges from 17% to almost 100% [38]. Autonomic symptoms, such as sweating, orthostatic hypotension, are one of the most common non-motor symptoms of PD”
Orthostatic hypotension could be a confounding factor. If it is present in early states of the disease is a red flag and the diagnosis is more an atypical parkinsonism (MSA most often). If it is present in advance PD is more a dose side effect of the dopamine and needs to be addressed with other medications. This is not considered as part of non-motor fluctuations.
7. PD continuously during daily lives have thusfar been rarely investigated. I don’t know if it was a typo or an idiom. Please correct.
8. The Methods are not clear for readers no familiarized with this topic, please make this section more readable.
9. The graphs seem to be disorganized and no mention of the meaning of each one in the text. Additionally, an explanation of each graph is worthy.
10. Predicting WO, defined as re-emergence of motor and non-motor symptoms that improve with further L-dopa intake [26], is a critical aspect of the clinical management of PD. Motor fluctuations can be observed in up to 50–80% of patients with advanced PD [35], and even in the early stage of disease, up to 50% of patients can develop motor fluctuation within the first 2 years of therapy [36].
Reference 36 is a comparation between Pramipexole and Dopamine as early treatment, and it states that : “our findings demonstrate that pramipexole, as initial therapy in patients with early PD, reduced the risk of developing prespecified dopaminergic motor complications by 55% compared with initiating therapy with levodopa over a 2-year period”.
I am having hard time finding anything to support that 50% of patients with PD can develop motor fluctuations within the first 2 years of therapy. It seems to be overrated because the first 5 years of treatment are called the “honeymoon” period since the response to levodopa is really good and support the diagnosis of PD. Please review and clarify.
11. In table 1 Patient number 3, the UPDRS was scored 0, which means patient was completely normal. Please review. Was the UPDRS in OFF or ON period?
Author Response
Dear Respected Reviewer,
We are thankful for your great comments. We did our best to answer them comprehensively. We hope you find the responses sufficient and proper. The added senteces to the manuscript are highlighted by green.
Best regards,
Authors.
Please find the point-to-point reply in the attachment.

Round 2
Reviewer 1 Report
Dear Authors
The changes made are satisfactory
Author Response
Dear Reviewer,
We appreciate that you find our responses satisfying. Thank you for the wonderful comments that helped us work on the manuscript and clarify our contribution.
Author Response
Dear Reviewer,
We address two of the remaining comments below.
Comment 10. Predicting WO, defined as re-emergence of motor and non-motor symptoms that improve with further L-dopa intake [26], is a critical aspect of the clinical management of PD. Motor fluctuations can be observed in up to 50– 80% of patients with advanced PD [35], and even in the early stage of disease, up to 50% of patients can develop motor fluctuation within the first 2 years of therapy [36]. Reference 36 is a comparison between Pramipexole and Dopamine as early treatment, and it states that : “our findings demonstrate that pramipexole, as initial therapy in patients with early PD, reduced the risk of developing prespecified dopaminergic motor complications by 55% compared with initiating therapy with levodopa over a 2- year period”. I am having a hard time finding anything to support that 50% of patients with PD can develop motor fluctuations within the first 2 years of therapy. It seems to be overrated because the first 5 years of treatment are called the “honeymoon” period since the response to levodopa is really good and supports the diagnosis of PD. Please review and clarify.
Response 10. We thank the reviewer for their opinion. Although in the results of the reference paper, this statement has not been explicitly stated, we note that in the same paper (Figure 2), there is a graph denoting the percentage of patients with motor fluctuations after starting levodopa or pramipexole following randomization. This graph denotes that almost 50% of patients developed wearing off by 700 days following randomization and initiation of levodopa. This is less than 2 years and hence, the statement that up to 50% of patients with PD can develop motor fluctuations within 2 years.
Reviewer response. Thank you so much for commenting on this point. As the authors mentioned the statement has not been explicitly stated in the paper and it is found in the graph. I am sure that as a reviewer I would have had comments to the author’s reference. I am still having hard time proving this sentence, but as the authors mentioned it is my experience and don’t have any good reference to support the opposite.
Authors’ second response. We thank the reviewer for the keen point regarding WO prevalence. Herein, we intended to highlight a study with a high prevalence among the many studies reporting the prevalence of WO, to emphasize its clinical relevance. A clinical article (Altavista et al, Parkinsonism Relat Disord. 2015, https://doi.org/10.1016/j.parkreldis.2014.11.002) about WO detection has also described that ‘WO phenomenon may occur after the first 2 years of treatment in up to 50% of patients’, referencing two papers including the same reference of ours (Stacy M et al.,Mov Disord, 2005, and Parkinson Study Group, JAMA, 2000). Thus, we believe that the epidemiologic basis of the sentence is reasonable.
Comment 11. In table 1 Patient number 3, the UPDRS was scored 0, which means patient was completely normal. Please review. Was the UPDRS in OFF or ON period?
Response 11. Thank you for this point. The UPDRS scores were assessed at the time of patients’ visit to the clinic. Thus they were not asked to withdraw medications, and would typically be considered in their “ON” state. Accordingly, we presume that patient #3 would have also been evaluated on his/her ON state. The key criteria for inclusion was that they had previously experienced frequent “OFF” episodes.
Reviewer response. Thank you so much for your response. In my experience I don’t recall any patient scored 0. It seems to be a person without PD, however understanding the inter-rater reliability, I defer this point to the research team.
Authors’ second response. It is true that healthy older subjects often score greater than 0. However, the patient had young-onset Parkinson’s disease (YOPD), and excellent response to levodopa with no or minimal symptoms in the ‘ON’ state.
